# Discrete Rényi Classifiers

**Meisam Razaviyayn**[*]
meisamr@stanford.edu

**Farzan Farnia**[*]
farnia@stanford.edu

**David Tse**[*]
dntse@stanford.edu

## Abstract

Consider the binary classification problem of predicting a target variable $Y$ from a discrete feature vector $\mathbf{X} = (X_1, \ldots, X_d)$. When the probability distribution $\mathbb{P}(\mathbf{X}, Y)$ is known, the optimal classifier, leading to the minimum misclassification rate, is given by the Maximum A-posteriori Probability (MAP) decision rule. However, in practice, estimating the complete joint distribution $\mathbb{P}(\mathbf{X}, Y)$ is computationally and statistically impossible for large values of $d$. Therefore, an alternative approach is to first estimate some low order marginals of the joint probability distribution $\mathbb{P}(\mathbf{X}, Y)$ and then design the classifier based on the estimated low order marginals. This approach is also helpful when the complete training data instances are not available due to privacy concerns.

In this work, we consider the problem of finding the optimum classifier based on some estimated low order marginals of $(\mathbf{X}, Y)$. We prove that for a given set of marginals, the minimum Hirschfeld-Gebelein-Rényi (HGR) correlation principle introduced in [1] leads to a *randomized* classification rule which is shown to have a misclassification rate no larger than twice the misclassification rate of the optimal classifier. Then, under a *separability* condition, it is shown that the proposed algorithm is equivalent to a randomized linear regression approach. In addition, this method naturally results in a robust feature selection method selecting a subset of features having the maximum worst case HGR correlation with the target variable. Our theoretical upper-bound is similar to the recent Discrete Chebyshev Classifier (DCC) approach [2], while the proposed algorithm has significant computational advantages since it only requires solving a least square optimization problem. Finally, we numerically compare our proposed algorithm with the DCC classifier and show that the proposed algorithm results in better misclassification rate over various UCI data repository datasets.

## 1 Introduction

Statistical classification, a core task in many modern data processing and prediction problems, is the problem of predicting labels for a given feature vector based on a set of training data instances containing feature vectors and their corresponding labels. From a probabilistic point of view, this problem can be formulated as follows: given data samples $(\mathbf{X}^1, Y^1), \ldots, (\mathbf{X}^n, Y^n)$ from a probability distribution $\mathbb{P}(\mathbf{X}, Y)$, predict the target label $y^{\text{test}}$ for a given test point $\mathbf{X} = \mathbf{x}^{\text{test}}$.

Many modern classification problems are on high dimensional *categorical* features. For example, in the genome-wide association studies (GWAS), the classification task is to predict a trait of interest based on observations of the SNPs in the genome. In this problem, the feature vector $\mathbf{X} = (X_1, \ldots, X_d)$ is categorical with $X_i \in \{0, 1, 2\}$.

What is the optimal classifier leading to the minimum misclassification rate for such a classification problem with high dimensional categorical feature vectors? When the joint probability distribution of the random vector $(\mathbf{X}, Y)$ is known, the MAP decision rule defined by $\delta^{\text{MAP}} \triangleq \operatorname{argmax}_y \mathbb{P}(Y =$

---

[*]Department of Electrical Engineering, Stanford University, Stanford, CA 94305.

$y|\mathbf{X} = \mathbf{x})$ achieves the minimum misclassification rate. However, in practice the joint probability distribution $\mathbb{P}(\mathbf{X}, Y)$ is not known. Moreover, estimating the complete joint probability distribution is not possible due to the curse of dimensionality. For example, in the above GWAS problem, the dimension of the feature vector $\mathbf{X}$ is $d \approx 3,000,000$ which leads to the alphabet size of $3^{3,000,000}$ for the feature vector $\mathbf{X}$. Hence, a practical approach is to first estimate some low order marginals of $\mathbb{P}(\mathbf{X}, Y)$, and then use these low order marginals to build a classifier with low misclassification rate. This approach, which is the sprit of various machine learning and statistical methods [2–6], is also useful when the complete data instances are not available due to privacy concerns in applications such as medical informatics.

In this work, we consider the above problem of building a classifier for a given set of low order marginals. First, we formally state the problem of finding the robust classifier with the minimum worst case misclassification rate. Our goal is to find a (possibly randomized) decision rule which has the minimum worst case misclassification rate over all probability distributions satisfying the given low order marginals. Then a surrogate objective function, which is obtained by the minimum HGR correlation principle [1], is used to propose a randomized classification rule. The proposed classification method has the worst case misclassification rate no more than twice the misclassification rate of the optimal classifier. When only pairwise marginals are estimated, it is shown that this classifier is indeed a randomized linear regression classifier on indicator variables under a *separability* condition. Then, we formulate a feature selection problem based on the knowledge of pairwise marginals which leads to the minimum misclassification rate. Our analysis provides a theoretical justification for using group lasso objective function for feature selection over the discrete set of features. Finally, we conclude by presenting numerical experiments comparing the proposed classifier with discrete Chebyshev classifier [2], Tree Augmented Naive Bayes [3], and Minimax Probabilistic Machine [4]. In short, the contributions of this work is as follows.

- Providing a rigorous theoretical justification for using the minimum HGR correlation principle for binary classification problem.

- Proposing a randomized classifier with misclassification rate no larger than twice the misclassification rate of the optimal classifier.

- Introducing a computationally efficient method for calculating the proposed randomized classifier when pairwise marginals are estimated and a *separability* condition is satisfied.

- Providing a mathematical justification based on maximal correlation for using group lasso problem for feature selection in categorical data.

**Related Work:** The idea of learning structures in data through low order marginals/moments is popular in machine learning and statistics. For example, the maximum entropy principle [7], which is the spirit of the variational method in graphical models [5] and tree augmented naive Bayes [3], is based on the idea of fixing the marginal distributions and fitting a probabilistic model which maximizes the Shannon entropy. Although these methods fit a probabilistic model satisfying the low order marginals, they do not directly optimize the misclassification rate of the resulting classifier.

Another related information theoretic approach is the minimum mutual information principle [8] which finds the probability distribution with the minimum mutual information between the feature vector and the target variable. This approach is closely related to the framework of this paper; however, unlike the minimum HGR principle, there is no known computationally efficient approach for calculating the probability distribution with the minimum mutual information.

In the continuous setting, the idea of minimizing the worst case misclassification rate leads to the *minimax probability machine* [4]. This algorithm and its analysis is not easily extendible to the discrete scenario.

The most related algorithm to this work is the recent *Discrete Chebyshev Classifier* (DCC) algorithm [2]. The DCC is based on the minimization of the worst case misclassification rate over the class of probability distributions with the given marginals of the form $(X_i, X_j, Y)$. Similar to our framework, the DCC method achieves the misclassification rate no larger than twice the misclassification rate of the optimum classifier. However, computation of the DCC classifier requires solving a non-separable non-smooth optimization problem which is computationally demanding, while the proposed algorithm results in a least squares optimization problem with a closed form solution. Furthermore, in contrast to [2] which only considers deterministic decision rules, in this work we

consider the class of randomized decision rules. Finally, it is worth noting that the algorithm in [2] requires *tree structure* to be tight, while our proposed algorithm works on non-tree structures as long as the separability condition is satisfied.

## 2 Problem Formulation

Consider the binary classification problem with $d$ discrete features $X_1, X_2, \ldots, X_d \in \mathcal{X}$ and a target variable $Y \in \mathcal{Y} \triangleq \{0, 1\}$. Without loss of generality, let us assume that $\mathcal{X} \triangleq \{1, 2, \ldots, m\}$ and the data points $(\mathbf{X}, Y)$ are coming from an underlying probability distribution $\bar{\mathbb{P}}_{\mathbf{X},Y}(\mathbf{x}, y)$. If the joint probability distribution $\bar{\mathbb{P}}(\mathbf{x}, y)$ is known, the optimal classifier is given by the maximum a posteriori probability (MAP) estimator, i.e., $\widehat{y}^{\mathrm{MAP}}(\mathbf{x}) \triangleq \mathrm{argmax}_{y \in \{0,1\}} \ \ \bar{\mathbb{P}}(Y = y \mid \mathbf{X} = \mathbf{x})$. However, the joint probability distribution $\bar{\mathbb{P}}(\mathbf{x}, y)$ is often not known in practice. Therefore, in order to utilize the MAP rule, one should first estimate $\bar{\mathbb{P}}(\mathbf{x}, y)$ using the training data instances. Unfortunately, estimating the joint probability distribution requires estimating the value of $\bar{\mathbb{P}}(\mathbf{X} = \mathbf{x}, Y = y)$ for all $(\mathbf{x}, y) \in \mathcal{X}^d \times \mathcal{Y}$ which is intractable for large values of $d$. Therefore, as mentioned earlier, our approach is to first estimate some low order marginals of the joint probability distribution $\bar{\mathbb{P}}(\cdot)$; and then utilize the minimax criterion for classification.

Let $\mathcal{C}$ be the class of probability distributions satisfying the estimated marginals. For example, when only pairwise marginals of the ground-truth distribution $\bar{\mathbb{P}}$ is estimated, the set $\mathcal{C}$ is the class of distributions satisfying the given pairwise marginals, i.e.,

$$\mathcal{C}_{\mathrm{pairwise}} \triangleq \left\{ \mathbb{P}_{\mathbf{X},Y}(\cdot, \cdot) \mid \mathbb{P}_{X_i,X_j}(x_i, x_j) = \bar{\mathbb{P}}_{X_i,X_j}(x_i, x_j), \ \mathbb{P}_{X_i,Y}(x_i, y) = \bar{\mathbb{P}}_{X_i,Y}(x_i, y), \right.$$
$$\left. \forall x_i, x_j \in \mathcal{X}, \ \forall y \in \mathcal{Y}, \ \forall i, j \right\}. \tag{1}$$

In general, $\mathcal{C}$ could be any class of probability distributions satisfying a set of estimated low order marginals.

Let us also define $\delta$ to be a randomized classification rule with

$$\delta(\mathbf{x}) = \begin{cases} 0 & \text{with probability } q_\delta^{\mathbf{x}} \\ 1 & \text{with probability } 1 - q_\delta^{\mathbf{x}}, \end{cases}$$

for some $q_\delta^{\mathbf{x}} \in [0, 1], \ \forall \mathbf{x} \in \mathcal{X}^d$. Given a randomized decision rule $\delta$ and a joint probability distribution $\mathbb{P}_{\mathbf{X},Y}(\mathbf{x}, y)$, we can extend $\mathbb{P}(\cdot)$ to include our randomized decision rule. Then the misclassification rate of the decision rule $\delta$, under the probability distribution $\mathbb{P}(\cdot)$, is given by $\mathbb{P}(\delta(\mathbf{X}) \neq Y)$. Hence, under minimax criterion, we are looking for a decision rule $\delta^*$ which minimizes the *worst case misclassification rate*. In other words, the robust decision rule is given by

$$\delta^* \in \underset{\delta \in \mathcal{D}}{\mathrm{argmin}} \ \underset{\mathbb{P} \in \mathcal{C}}{\max} \ \mathbb{P}\left(\delta(\mathbf{X}) \neq Y\right), \tag{2}$$

where $\mathcal{D}$ is the set of all randomized decision rules. Notice that the optimal decision rule $\delta^*$ may not be unique in general.

## 3 Worst Case Error Minimization

In this section, we propose a surrogate objective for (2) which leads to a decision rule with misclassification rate no larger than twice of the optimal decision rule $\delta^*$. Later we show that the proposed surrogate objective is connected to the minimum HGR principle [1].

Let us start by rewriting (2) as an optimization problem over real valued variables. Notice that each probability distribution $\mathbb{P}_{\mathbf{X},Y}(\cdot, \cdot)$ can be represented by a probability vector $\mathbf{p} = [p_{\mathbf{x},y} \mid (\mathbf{x}, y) \in \mathcal{X}^d \times \mathcal{Y}] \in \mathbb{R}^{2m^d}$ with $p_{\mathbf{x},y} = \mathbb{P}(\mathbf{X} = \mathbf{x}, Y = y)$ and $\sum_{\mathbf{x},y} p_{\mathbf{x},y} = 1$. Similarly, every randomized rule $\delta$ can be represented by a vector $\mathbf{q}_\delta = [q_\delta^{\mathbf{x}} \mid \mathbf{x} \in \mathcal{X}^d] \in \mathbb{R}^{m^d}$. Adopting these notations, the set $\mathcal{C}$ can be rewritten in terms of the probability vector $\mathbf{p}$ as

$$\mathcal{C} \triangleq \left\{ \mathbf{p} \mid \mathbf{A}\mathbf{p} = \mathbf{b}, \ \mathbf{1}^T \mathbf{p} = 1, \ \mathbf{p} \geq \mathbf{0} \right\},$$

where the system of linear equations $\mathbf{A}\mathbf{p} = \mathbf{b}$ represents all the low order marginal constraints in $\mathcal{B}$; and the notation $\mathbf{1}$ denotes the vector of all ones. Therefore, problem (2) can be reformulated as

$$\mathbf{q}_\delta^* \in \underset{\mathbf{0} \leq \mathbf{q}_\delta \leq \mathbf{1}}{\operatorname{argmin}} \ \max_{\mathbf{p} \in \mathcal{C}} \sum_{\mathbf{x}} \left( q_\delta^{\mathbf{x}} p_{\mathbf{x},1} + (1 - q_\delta^{\mathbf{x}}) p_{\mathbf{x},0} \right), \tag{3}$$

where $p_{x,0}$ and $p_{x,1}$ denote the elements of the vector $\mathbf{p}$ corresponding to the probability values $\mathbb{P}(\mathbf{X} = \mathbf{x}, Y = 0)$ and $\mathbb{P}(\mathbf{X} = \mathbf{x}, Y = 1)$, respectively. The simple application of the minimax theorem [9] implies that the saddle point of the above optimization problem exists and moreover, the optimal decision rule is a MAP rule for a certain probability distribution $\mathbb{P}^* \in \mathcal{C}$. In other words, there exists a pair $(\delta^*, \mathbb{P}^*)$ for which

$$\mathbb{P}(\delta^*(\mathbf{X}) \neq Y) \leq \mathbb{P}^*(\delta^*(\mathbf{X}) \neq Y), \ \forall \mathbb{P} \in \mathcal{C} \ \text{ and } \ \mathbb{P}^*(\delta(\mathbf{X}) \neq Y) \geq \mathbb{P}^*(\delta^*(\mathbf{X}) \neq Y), \ \forall \delta \in \mathcal{D}.$$

Although the above observation characterizes the optimal decision rule to some extent, it does not provide a computationally efficient approach for finding the optimal decision rule. Notice that it is NP-hard to verify the existence of a probability distribution satisfying a given set of low order marginals [10]. Based on this observation and the result in [11], we conjecture that in general, solving (2) is NP-hard in the number variables and the alphabet size even when the set $\mathcal{C}$ is non-empty. Hence, here we focus on developing a framework to find an *approximate* solution of (2).

Let us continue by utilizing the minimax theorem [9] and obtain the worst case probability distribution in (3) by $\mathbf{p}^* \in \operatorname{argmax}_{\mathbf{p} \in \mathcal{C}} \min_{\mathbf{0} \leq \mathbf{q}_\delta \leq \mathbf{1}} \sum_{\mathbf{x}} \left( \mathbf{q}_\delta^{\mathbf{x}} \mathbf{p}_{\mathbf{x},1} + (1 - \mathbf{q}_\delta^{\mathbf{x}}) p_{\mathbf{x},0} \right)$, or equivalently,

$$\mathbf{p}^* \in \underset{\mathbf{p} \in \mathcal{C}}{\operatorname{argmax}} \sum_{\mathbf{x}} \min \left\{ p_{\mathbf{x},0} \ , \ p_{\mathbf{x},1} \right\}. \tag{4}$$

Despite convexity of the above problem, there are two sources of hardness which make the problem intractable for moderate and large values of $d$. Firstly, the objective function is non-smooth. Secondly, the number of optimization variables is $2m^d$ and grows exponentially with the alphabet size. To deal with the first issue, notice that the function inside the summation is the max-min fairness objective between the two quantities $p_{\mathbf{x},1}$ and $p_{\mathbf{x},0}$. Replacing this objective with the harmonic average leads to the following smooth convex optimization problem:

$$\widetilde{\mathbf{p}} \in \underset{\mathbf{p} \in \mathcal{C}}{\operatorname{argmax}} \sum_{\mathbf{x}} \frac{p_{\mathbf{x},1} p_{\mathbf{x},0}}{p_{\mathbf{x},1} + p_{\mathbf{x},0}}. \tag{5}$$

It is worth noting that the harmonic mean of the two quantities is intuitively a reasonable surrogate for the original objective function since

$$\frac{p_{\mathbf{x},1} p_{\mathbf{x},0}}{p_{\mathbf{x},1} + p_{\mathbf{x},0}} \leq \min \left\{ p_{\mathbf{x},0} \ , \ p_{\mathbf{x},1} \right\} \leq \frac{2 p_{\mathbf{x},1} p_{\mathbf{x},0}}{p_{\mathbf{x},1} + p_{\mathbf{x},0}}. \tag{6}$$

Although this inequality suggests that the objective functions in (5) and (4) are close to each other, it is not clear whether the distribution $\widetilde{\mathbf{p}}$ leads to any classification rule having low misclassification rate for *all* distributions in $\mathcal{C}$. In order to obtain a classification rule from $\widetilde{\mathbf{p}}$, the first naive approach is to use MAP decision rule based on $\widetilde{\mathbf{p}}$. However, the following result shows that this decision rule does not achieve the factor two misclassification rate obtained in [2].

**Theorem 1** *Let us define* $\widetilde{\delta}^{\mathrm{map}}(\mathbf{x}) \triangleq \operatorname{argmax}_{y \in \mathcal{Y}} \widetilde{\mathbf{p}}_{\mathbf{x},y}$ *with the worst case error probability* $\widetilde{e}^{\mathrm{map}} \triangleq \max_{\mathbb{P} \in \mathcal{C}} \ \mathbb{P}\left( \widetilde{\delta}^{\mathrm{map}}(\mathbf{X}) \neq Y \right)$. *Then,* $e^* \leq \widetilde{e}^{\mathrm{map}} \leq 4e^*$, *where* $e^*$ *is the worst case misclassification rate of the optimal decision rule* $\delta^*$, *that is,* $e^* \triangleq \max_{\mathbb{P} \in \mathcal{C}} \ \mathbb{P}\left( \delta^*(\mathbf{X}) \neq Y \right)$.

**Proof** The proof is similar to the proof of next theorem and hence omitted here.

Next we show that, surprisingly, one can obtain a randomized decision rule based on the solution of (5) which has a misclassification rate no larger than twice of the optimal decision rule $\delta^*$.

Given $\widetilde{\mathbf{p}}$ as the optimal solution of (5), define the random decision rule $\widetilde{\delta}$ as

$$\widetilde{\delta}(\mathbf{x}) = \begin{cases} 0 & \text{with probability } \frac{\widetilde{p}_{\mathbf{x},0}^2}{\widetilde{p}_{\mathbf{x},0}^2 + \widetilde{p}_{\mathbf{x},1}^2} \\ 1 & \text{with probability } \frac{\widetilde{p}_{\mathbf{x},1}^2}{\widetilde{p}_{\mathbf{x},0}^2 + \widetilde{p}_{\mathbf{x},1}^2} \end{cases} \tag{7}$$

Let $\tilde{e}$ be the worst case classification error of the decision rule $\tilde{\delta}$, i.e.,

$$\tilde{e} \triangleq \max_{\mathbb{P} \in \mathcal{C}} \mathbb{P}\left(\tilde{\delta}(\mathbf{X}) \neq Y\right).$$

Clearly, $e^* \leq \tilde{e}$ according to the definition of the optimal decision rule $e^*$. The following theorem shows that $\tilde{e}$ is also upper-bounded by twice of the optimal misclassification rate $e^*$.

**Theorem 2** *Define*

$$\theta \triangleq \max_{\mathbf{p} \in \mathcal{C}} \sum_{\mathbf{x}} \frac{p_{\mathbf{x},1} p_{\mathbf{x},0}}{p_{\mathbf{x},1} + p_{\mathbf{x},0}} \tag{8}$$

*Then, $\theta \leq \tilde{e} \leq 2\theta \leq 2e^*$. In other words, the worst case misclassification rate of the decision rule $\tilde{\delta}$ is at most twice the optimal decision rule $\delta^*$.*

**Proof** The proof is relegated to the supplementary materials.

So far, we have resolved the non-smoothness issue in solving (4) by using a surrogate objective function. In the next section, we resolve the second issue by establishing the connection between problem (5) and the minimum HGR correlation principle [1]. Then, we use the existing result in [1] to develop a computationally efficient approach for calculating the decision rule $\tilde{\delta}(\cdot)$ for $\mathcal{C}_{\text{pairwise}}$.

## 4    Connection to Hirschfeld-Gebelein-Rényi Correlation

A commonplace approach to infer models from data is to employ the maximum entropy principle [7]. This principle states that, given a set of constraints on the ground-truth distribution, the distribution with the maximum (Shannon) entropy under those constraints is a proper representer of the class. To extend this rule to the classification problem, the authors in [8] suggest to pick the distribution maximizing the target entropy conditioned to features, or equivalently minimizing mutual information between target and features. Unfortunately, this approach does not lead to a computationally efficient approach for model fitting and there is no guarantee on the misclassification rate of the resulting classifier. Here we study an alternative approach of minimum HGR correlation principle [1]. This principle suggests to pick the distribution in $\mathcal{C}$ minimizing HGR correlation between the target variable and features. The HGR correlation coefficient between the two random objects $\mathbf{X}$ and $Y$, which was first introduced by Hirschfeld and Gebelein [12, 13] and then studied by Rényi [14], is defined as $\rho(\mathbf{X}, Y) \triangleq \sup_{f,g} \mathbb{E}\left[f(\mathbf{X}) g(Y)\right]$, where the maximization is taken over the class of all measurable functions $f(\cdot)$ and $g(\cdot)$ with $\mathbb{E}[f(X)] = \mathbb{E}[g(Y)] = 0$ and $\mathbb{E}[f^2(\mathbf{X})] = \mathbb{E}[g^2(Y)] = 1$. The HGR correlation coefficient has many desirable properties. For example, it is normalized to be between $0$ and $1$. Furthermore, this coefficient is zero if and only if the two random variables are independent; and it is one if there is a strict dependence between $\mathbf{X}$ and $Y$. For other properties of the HGR correlation coefficient see [14, 15] and the references therein.

**Lemma 1** *Assume the random variable $Y$ is binary and define $q \triangleq \mathbb{P}(Y = 0)$. Then,*

$$\rho(\mathbf{X}, Y) = \sqrt{1 - \frac{1}{q(1-q)} \sum_{\mathbf{x}} \left[ \frac{\mathbb{P}_{\mathbf{X},Y}(\mathbf{x}, 0) \mathbb{P}_{\mathbf{X},Y}(\mathbf{x}, 1)}{\mathbb{P}_{\mathbf{X},Y}(\mathbf{x}, 0) + \mathbb{P}_{\mathbf{X},Y}(\mathbf{x}, 1)} \right]},$$

**Proof** The proof is relegated to the supplementary material.

This lemma leads to the following observation.

**Observation:** Assume the marginal distribution $\mathbb{P}(Y = 0)$ and $\mathbb{P}(Y = 1)$ is fixed for any distribution $\mathbb{P} \in \mathcal{C}$. Then, the distribution in $\mathcal{C}$ with the minimum HGR correlation between $\mathbf{X}$ and $Y$ is the distribution $\tilde{\mathbb{P}}$ obtained by solving (5). In other words, $\rho(\mathbf{X}, Y; \tilde{\mathbb{P}}) \leq \rho(\mathbf{X}, Y; \mathbb{P})$, $\forall \, \mathbb{P} \in \mathcal{C}$, where $\rho(\mathbf{X}, Y; \mathbb{P})$ denotes the HGR correlation coefficient under the probability distribution $\mathbb{P}$.

Based on the above observation, from now on, we call the classifier $\tilde{\delta}(\cdot)$ in (7) as the *"Rényi classifier"*. In the next section, we use the result of the recent work [1] to compute the Rényi classifier $\tilde{\delta}(\cdot)$ for a special class of marginals $\mathcal{C} = \mathcal{C}_{\text{pairwise}}$.

# 5 Computing Rényi Classifier Based on Pairwise Marginals

In many practical problems, the number of features $d$ is large and therefore, it is only computationally tractable to estimate marginals of order at most two. Hence, hereafter, we restrict ourselves to the case where only the first and second order marginals of the distribution $\bar{\mathbb{P}}$ is estimated, i.e., $\mathcal{C} = \mathcal{C}_{\text{pairwise}}$. In this scenario, in order to predict the output of the Rényi classifier for a given data point $\mathbf{x}$, one needs to find the value of $\widetilde{\mathbf{p}}_{\mathbf{x},0}$ and $\widetilde{\mathbf{p}}_{\mathbf{x},1}$. Next, we state a result from [1] which sheds light on the computation of $\widetilde{\mathbf{p}}_{\mathbf{x},0}$ and $\widetilde{\mathbf{p}}_{\mathbf{x},1}$. To state the theorem, we need the following definitions:

Let the matrix $\mathbf{Q} \in \mathbb{R}^{dm \times dm}$ and the vector $\mathbf{d} \in \mathbb{R}^{dm \times 1}$ be defined through their entries as

$$\mathbf{Q}_{mi+k,mj+\ell} = \bar{\mathbb{P}}(X_{i+1} = k, X_{j+1} = \ell), \quad \mathbf{d}_{mi+k} = \bar{\mathbb{P}}(X_{i+1} = k, Y = 1) - \bar{\mathbb{P}}(X_{i+1} = k, Y = 0),$$

for every $i, j = 0, \ldots, d-1$ and $k, \ell = 1, \ldots, m$. Also define the function $h(\mathbf{z}) : \mathbb{R}^{md \times 1} \mapsto \mathbb{R}$ as $h(\mathbf{z}) \triangleq \sum_{i=1}^{d} \max\{\mathbf{z}_{mi-m+1}, \mathbf{z}_{mi-m+2}, \ldots, \mathbf{z}_{mi}\}$. Then, we have the following theorem.

**Theorem 3** *(Rephrased from [1]) Assume $\mathcal{C}_{\text{pairwise}} \neq \emptyset$. Let*

$$\gamma \triangleq \min_{\mathbf{z} \in \mathbb{R}^{md \times 1}} \mathbf{z}^T \mathbf{Q} \mathbf{z} - \mathbf{d}^T \mathbf{z} + \frac{1}{4}. \tag{9}$$

*Then, $\sqrt{1 - \frac{\gamma}{q(1-q)}} \leq \min_{\mathbb{P} \in \mathcal{C}_{\text{pairwise}}} \rho(\mathbf{X}, Y; \mathbb{P})$, where the inequality holds with equality if and only if there exists a solution $\mathbf{z}^*$ to (9) such that $h(\mathbf{z}^*) \leq \frac{1}{2}$ and $h(-\mathbf{z}^*) \leq \frac{1}{2}$; or equivalently, if and only if the following separability condition is satisfied for some $\mathbb{P} \in \mathcal{C}_{\text{pairwise}}$.*

$$\mathbb{E}_{\mathbb{P}}[Y | \mathbf{X} = \mathbf{x}] = \sum_{i=1}^{d} \zeta_i(x_i), \quad \forall \mathbf{x} \in \mathcal{X}^d, \tag{10}$$

*for some functions $\zeta_1, \ldots, \zeta_d$. Moreover, if the separability condition holds with equality, then*

$$\widetilde{\mathbb{P}}(Y = y | \mathbf{X} = (x_1, \ldots, x_d)) = \frac{1}{2} - (-1)^y \sum_{i=1}^{d} z^*_{(i-1)m+x_i}. \tag{11}$$

Combining the above theorem with the equality

$$\frac{\widetilde{\mathbb{P}}^2(Y = 0, \mathbf{X} = \mathbf{x})}{\widetilde{\mathbb{P}}^2(Y = 0, \mathbf{X} = \mathbf{x}) + \widetilde{\mathbb{P}}^2(Y = 1, \mathbf{X} = \mathbf{x})} = \frac{\widetilde{\mathbb{P}}^2(Y = 0 | \mathbf{X} = \mathbf{x})}{\widetilde{\mathbb{P}}^2(Y = 0 | \mathbf{X} = \mathbf{x}) + \widetilde{\mathbb{P}}^2(Y = 1 | \mathbf{X} = \mathbf{x})}$$

implies that the decision rule $\widetilde{\delta}$ and $\widetilde{\delta}^{\text{map}}$ can be computed in a computationally efficient manner under the separability condition. Notice that when the separability condition is not satisfied, the approach proposed in this section would provide a classification rule whose error rate is still bounded by $2\gamma$. However, this error rate does no longer provide a 2-factor approximation gap. It is also worth mentioning that the separability condition is a property of the class of distribution $\mathcal{C}_{\text{pairwise}}$ and is independent of the classifier at hand. Moreover, this condition is satisfied with a positive measure over the simplex of the all probability distributions, as discussed in [1]. Two remarks are in order:

**Inexact knowledge of marginal distribution:** The optimization problem (9) is equivalent to solving the stochastic optimization problem

$$\mathbf{z}^* = \underset{\mathbf{z}}{\operatorname{argmin}} \, \mathbb{E}\left[\left(\mathbf{W}^T \mathbf{z} - C\right)^2\right],$$

where $\mathbf{W} \in \{0, 1\}^{md \times 1}$ is a random vector with $\mathbf{W}_{m(i-1)+k} = 1$ if $X_i = k$ in the and $\mathbf{W}_{m(i-1)+k} = 0$, otherwise. Also define the random variable $C \in \{-\frac{1}{2}, \frac{1}{2}\}$ with $C = \frac{1}{2}$ if the random variable $Y = 1$ and $C = -\frac{1}{2}$, otherwise. Here the expectation could be calculated with respect to any distribution in $\mathcal{C}$. Hence, in practice, the above optimization problem can be estimated using Sample Average Approximation (SAA) method [16, 17] through the optimization problem

$$\widehat{\mathbf{z}} = \underset{\mathbf{z}}{\operatorname{argmin}} \, \frac{1}{n} \sum_{i=1}^{n} \left((\mathbf{w}^i)^T \mathbf{z} - c^i\right)^2,$$

where $(\mathbf{w}^i, c^i)$ corresponds to the $i$-th training data point $(\mathbf{x}^i, y^i)$. Clearly, this is a least square problem with a closed form solution. Notice that in order to bound the SAA error and avoid overfitting, one could restrict the search space for $\widehat{\mathbf{z}}$ [18]. This could also be done using regularizers such as ridge regression by solving

$$\widehat{\mathbf{z}}^{\text{ridge}} = \operatorname*{argmin}_{\mathbf{z}} \frac{1}{n} \sum_{i=1}^{n} \left( (\mathbf{w}^i)^T \mathbf{z} - c^i \right)^2 + \lambda^{\text{ridge}} \|\mathbf{z}\|_2^2.$$

**Beyond pairwise marginals:** When $d$ is small, one might be interested in estimating higher order marginals for predicting $Y$. In this scenario, a simple modification for the algorithm is to define the new set of feature random variables $\left\{ \widetilde{X}_{ij} = (X_i, X_j) \mid i \neq j \right\}$; and apply the algorithm to the new set of feature variables. It is not hard to see that this approach utilizes the marginal information $\mathbb{P}(X_i, X_j, X_k, X_\ell)$ and $\mathbb{P}(X_i, X_j, Y)$.

# 6 Robust Rényi Feature Selection

The task of feature selection for classification purposes is to preselect a subset of features for use in model fitting in prediction. Shannon mutual information, which is a measure of dependence between two random variables, is used in many recent works as an objective for feature selection [19, 20]. In these works, the idea is to select a small subset of features with maximum dependence with the target variable $Y$. In other words, the task is to find a subset of variables $\mathcal{S} \subseteq \{1, \ldots, d\}$ with $|\mathcal{S}| \leq k$ based on the following optimization problem

$$\mathcal{S}^{\text{MI}} \triangleq \operatorname*{argmax}_{\mathcal{S} \subseteq \{1, \ldots, d\}} \mathcal{I}(\mathbf{X}_{\mathcal{S}}; Y), \tag{12}$$

where $\mathbf{X}_{\mathcal{S}} \triangleq (X_i)_{i \in \mathcal{S}}$ and $\mathcal{I}(\mathbf{X}_{\mathcal{S}}; Y)$ denotes the mutual information between the random variable $\mathbf{X}_{\mathcal{S}}$ and $Y$. Almost all of the existing approaches for solving (12) are based on heuristic approaches and of greedy nature which aim to find a *sub-optimal* solution of (12). Here, we suggest to replace mutual information with the maximal correlation. Furthermore, since estimating the joint distribution of $\mathbf{X}$ and $Y$ is computationally and statistically impossible for large number of features $d$, we suggest to estimate some low order marginals of the groundtruth distribution $\bar{\mathbb{P}}(\mathbf{X}, Y)$ and then solve the following *robust Rényi feature selection* problem:

$$\mathcal{S}^{\text{RFS}} \triangleq \operatorname*{argmax}_{\mathcal{S} \subseteq \{1, \ldots, d\}} \min_{\mathbb{P} \in \mathcal{C}} \rho(\mathbf{X}_{\mathcal{S}}, Y; \mathbb{P}). \tag{13}$$

When only pairwise marginals are estimated from the training data, i.e., $\mathcal{C} = \mathcal{C}_{\text{pairwise}}$, maximizing the lower-bound $\sqrt{1 - \frac{\gamma}{q(1-q)}}$ instead of (13) leads to the following optimization problem

$$\widehat{\mathcal{S}}^{\text{RFS}} \triangleq \operatorname*{argmax}_{|\mathcal{S}| \leq k} \sqrt{1 - \frac{1}{q(1-q)} \min_{\mathbf{z} \in \mathcal{Z}_{\mathcal{S}}} \mathbf{z}^T \mathbf{Q} \mathbf{z} - \mathbf{d}^T \mathbf{z} + \frac{1}{4}},$$

or equivalently,

$$\widehat{\mathcal{S}}^{\text{RFS}} \triangleq \operatorname*{argmin}_{|\mathcal{S}| \leq k} \min_{\mathbf{z} \in \mathcal{Z}_{\mathcal{S}}} \mathbf{z}^T \mathbf{Q} \mathbf{z} - \mathbf{d}^T \mathbf{z},$$

where $\mathcal{Z}_{\mathcal{S}} \triangleq \left\{ \mathbf{z} \in \mathbb{R}^{md} \mid \sum_{k=1}^{m} |z_{mi-m+k}| = 0, \, \forall i \notin \mathcal{S} \right\}$. This problem is of combinatorial nature. Howevre, using the standard group Lasso regularizer leads to the feature selection procedure in Algorithm 1.

---

**Algorithm 1** Robust Rényi Feature Selection

---

Choose a regularization parameter $\lambda > 0$ and define $h(\mathbf{z}) \triangleq \sum_{i=1}^{d} \max\{z_{mi-m+1}, \ldots, z_{mi}\}$.

Let $\quad \widehat{\mathbf{z}}^{\text{RFS}} \in \operatorname*{argmin}_{\mathbf{z}} \mathbf{z}^T \mathbf{Q} \mathbf{z} - \mathbf{d}^T \mathbf{z} + \lambda h(|\mathbf{z}|)$.

Set $\quad \mathcal{S} = \{i \mid \sum_{k=1}^{m} |z_{mi-m+k}^{\text{RFS}}| > 0\}$.

---

Notice that, when the pairwise marginals are estimated from a set of training data points, the above feature selection procedure is equivalent to applying the group Lasso regularizer to the standard linear regression problem over the domain of indicator variables. Our framework provides a justification for this approach based on the robust maximal correlation feature selection problem (13).

**Remark 1** *Another natural approach to define the feature selection procedure is to select a subset of features $\mathcal{S}$ by minimizing the worst case classification error, i.e., solving the following optimization problem*

$$\min_{|\mathcal{S}| \leq k} \min_{\delta \in \mathcal{D}_{\mathcal{S}}} \max_{\mathbb{P} \in \mathcal{C}} \mathbb{P}(\delta(\mathbf{X}) \neq Y), \tag{14}$$

*where $\mathcal{D}_{\mathcal{S}}$ is the set of randomized decision rules which only uses the feature variables in $\mathcal{S}$. Define $\mathcal{F}(\mathcal{S}) \triangleq \min_{\delta \in \mathcal{D}_{\mathcal{S}}} \max_{\mathbb{P} \in \mathcal{C}} \mathbb{P}(\delta(\mathbf{X}) \neq Y)$. It can be shown that $\mathcal{F}(\mathcal{S}) \leq \min_{|\mathcal{S}| \leq k} \min_{\mathbf{z} \in \mathcal{Z}_{\mathcal{S}}} \mathbf{z}^T \mathbf{Q} \mathbf{z} - \mathbf{d}^T \mathbf{z} + \frac{1}{4}$. Therefore, another justification for Algorithm 1 is to minimize an upper-bound of $\mathcal{F}(\mathcal{S})$ instead of itself.*

**Remark 2** *Alternating Direction Method of Multipliers (ADMM) algorithm [21] can be used for solving the optimization problem in Algorithm 1; see the supplementary material for more details.*

## 7 Numerical Results

We evaluated the performance of the Rényi classifiers $\widetilde{\delta}$ and $\widetilde{\delta}^{\mathrm{map}}$ on five different binary classification datasets from the UCI machine learning data repository. The results are compared with five different benchmarks used in [2]: Discrete Chebyshev Classifier [2], greedy DCC [2], Tree Augmented Naive Bayes [3], Minimax Probabilistic Machine [4], and support vector machines (SVM). In addition to the classifiers $\widetilde{\delta}$ and $\widetilde{\delta}^{\mathrm{map}}$ which only use pairwise marginals, we also use higher order marginals in $\widetilde{\delta}_2$ and $\widetilde{\delta}_2^{\mathrm{map}}$. These classifiers are obtained by defining the new feature variables $\{\widetilde{X}_{ij} = (X_i, X_j)\}$ as discussed in section 5. Since in this scenario, the number of features is large, we combine our Rényi classifier with the proposed group lasso feature selection. In other words, we first select a subset of $\{\widetilde{X}_{ij}\}$ and then find the maximum correlation classifier for the selected features. The value of $\lambda^{\mathrm{ridge}}$ and $\lambda$ is determined through cross validation. The results are averaged over 100 Monte Carlo runs each using 70% of the data for training and the rest for testing. The results are summarized in the table below where each number shows the percentage of the error of each method. The boldface numbers denote the best performance on each dataset.

As can be seen in this table, in four of the tested datasets, at least one of the proposed methods outperforms the other benchmarks. Furthermore, it can be seen that the classifier $\widetilde{\delta}^{\mathrm{map}}$ on average performs better than $\widetilde{\delta}$. This fact could be due to the specific properties of the underlying probability distribution in each dataset.

| Datasets | $\widetilde{\delta}^{\mathrm{map}}$ | $\widetilde{\delta}$ | $\widetilde{\delta}_2^{\mathrm{map}}$ | $\widetilde{\delta}_2$ | $\widetilde{\delta}_{\mathrm{FS},2}^{\mathrm{map}}$ | $\widetilde{\delta}_{\mathrm{FS},2}$ | DCC | gDCC | MPM | TAN | SVM |
|---|---|---|---|---|---|---|---|---|---|---|---|
| adult | 17 | 21 | **16** | 20 | **16** | 20 | 18 | 18 | 22 | 18 | 22 |
| credit | **13** | 16 | 16 | 17 | 16 | 17 | 14 | **13** | **13** | 17 | 16 |
| kr-vs-kp | 5 | 10 | 5 | 14 | 5 | 14 | 10 | 10 | 5 | 7 | **3** |
| promoters | 6 | 16 | **3** | 4 | **3** | 4 | 5 | **3** | 6 | 44 | 9 |
| votes | 3 | 4 | 3 | 4 | **2** | 4 | 3 | 3 | 4 | 8 | 5 |

In order to evaluate the computational efficiency of the Rényi classifier, we compare its running time with SVM over the synthetic data set with $d = 10,000$ features and $n = 200$ data points. Each feature $X_i$ is generated by i.i.d. Bernoulli distribution with $\mathbb{P}(X_i = 1) = 0.7$. The target variable $y$ is generated by $y = \mathrm{sign}(\alpha^T \mathbf{X} + n)$ with $n \sim \mathcal{N}(0,1)$; and $\alpha \in \mathbb{R}^d$ is generated with 30% nonzero elements each drawn from standard Gaussian distribution $\mathcal{N}(0,1)$. The results are averaged over 1000 Monte-Carlo runs of generating the data set and use 85% of the data points for training and 15% for test. The Rényi classifier is obtained by gradient descent method with regularizer $\lambda^{\mathrm{ridge}} = 10^4$. The numerical experiment shows 19.7% average misclassification rate for SVM and 19.9% for Rényi classifier. However, the average training time of the Rényi classifier is 0.2 seconds while the training time of SVM (with Matlab SVM command) is 1.25 seconds.

**Acknowledgments:** The authors are grateful to Stanford University supporting a Stanford Graduate Fellowship, and the Center for Science of Information (CSoI), an NSF Science and Technology Center under grant agreement CCF-0939370 , for the support during this research.

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
