[Reviews · NeurIPS 2015]

Submitted by Assigned_Reviewer_1

This paper studies the learning of optimal classifiers regarding discrete features on some estimated low order marginals of (X, y). Directly learning optimal classifiers for data with discrete features based on various marginals is interesting, like [Eban, ICML14].

The proposed classifier learning model is the optimization problem in (9), which is interesting. But I found it should be very expensive since Q \in dm\times dm, especially when doing feature selection where a closed-form solution does not exist.

Moreover, when constructing Q, how to compute \bar{P(X_i=k, X_j=l)}? Is it efficient? Is Q necessarily positive-definite?

When learning classifiers, the support vector machine (SVM) is a well-known method. Are there any drawbacks for SVM to learn a classifier regarding discrete features? In the comparison, the author did not include SVM for comparison, which is very strange.

The optimization problem in (13) or the problem below is a non-convex problem, thus should be very hard to solve. The paper proposes to use a (group) LASSO regularization to achieve feature selection. However, there should be a big gap between the optimization problem and the algorithm. The paper says that the gap is 0. I do not think so. At least, I do not find detailed discussions about this in the paper.

In the experiments, the details of data sets are not given.

Summary: This paper studies the learning of optimal classifiers regarding discrete features on some estimated low order marginals of (X, y). The proposed formulation could be very expensive for high-dimensional cases.

Submitted by Assigned_Reviewer_2

The connection and flow of the theoretical sections is good. Theorem 3 and the formulation and solving of (9) are very interesting.

The experiments could be done better and provide better confidence in the method. The current results do not show significant improvement to other methods.

Details:

Can you provide some intuition of the separability condition in theorem 3? Is it related to the traditional separability condition in linear classification ?

Typo: Line 329 extra "in the"

Experimental results:

Do you think this method will be much better than a naive linear classifier (e.g SVM) that use a mxd weight vector W with w(i,j) = weight of value i of feature j? Perhaps that could be a baseline.

The motivation comes from high dimensional data where full distribution is infeasible to estimate from limited training data. Authors mention GWAS data at the beginning of the paper. It would be much better to have some datasets that could be used to demonstrate the power of the method in very high dimension setting.

Summary: The paper introduces a classifier for discrete features based on a given set of low order marginals. The paper is well written and

theoretical result is clearly presented.

Although the main theorem is inspired by another paper ([1]), applying it to classification problems to me is novel. My main concern is the experimental result is relatively weak compared to a well presented theoretical sections.

Submitted by Assigned_Reviewer_3

Quality .... The paper was interesting to read and contained many nice ideas, intuitions, and theoretical connections. The technical contributions in Sections 3-5 are sound and clever, although in some cases building on previous works [2,4]. Moreover, the problem formulation as a linear optimization and the least squared solution make the approach computationally favorable. However, there are points that need more clarification/elaboration in the paper too, in order to better judge the practical application and impact of the work. For instance, it's never discussed where the set of low order marginal come from. Also, the required separability condition for the theoretical bound on the error should be more elaborated, e.g., how does it relate to a method like naive bayes?. The empirical evaluation shows that the proposed approach can compete and in cases outperform the competition, however the difference in performances is not quite pronounced, and the intuition performance of the proposed mehtod is not really provided (mostly relegated to a future work).

Clarity ..... The paper is overall well-written and easy to read. I typically found the developments intriguing and clear. Many nice theoretical connections are made through the paper, which help to understand better the contributions (and sometimes other works, e.g., group Lasso).

Originality ..... The work builds on previous work in the approach [4] and the theoretical guarantees it aims at [2], however, I believe the paper still makes sufficient, novel theoretical advancement, insights and intuitions. The technical developments and the connections made in Sections 3-5 are in specific interesting novel contributions.

Significance ..... I believe the paper makes many interesting theoretical contributions to the problem domain, which are worthwhile. It also includes clever formulations and solutions, which make it interesting to read. However, I am not fully convinced of the practical implications of the work, mainly the limitations of the results (the separability condition, the set of given pairwise marginals), and the empirical evaluation, which is not very extensive.

Summary: The paper presents a method for inference (prediction) given a set of (low-order/pairwise) marginals. It derives a randomized classifier, which is shown to have no worse than twice the optimal error rate (similar to [2]), and presents a theoretical connection to the HGR correlation. The empirical results are reasonable, but not quite extensive, and not depicting great improvements. While I like the theoretical developments of the paper, which are clearly presented, I am not quite convinced of the significance of the contributions, and the impact they can make, in part due to doubts about the applicability/limitations of the assumptions/settings behind the result.

Submitted by Assigned_Reviewer_4

The problem of fitting a classifier to some data is investigated in this paper. Since full join distributions are infeasible when the number of features is large, the authors follow a standard approach of using marginal distributions to build a classifier. The extreme example of this approach is a Naive Bayes classifier which assumes that all features are independent given the class, and marginal for individual features only are considered. In this work, having pairwise marginal distributions the authors search for stochastic decision rules which minimise error for the worst case joint probability consistent with marginal distributions; those marginal distributions are computed from the data.

The authors could also mention that even if the full joint could be modelled, it is not always wise to do that because it usually leads to over-fitting as, e.g., in unbiased Bayesian learning. But, that's a rather minor thing.

The paper is very well written, and it contains several, important, technical contributions. The discussion of related work seems to be sufficient. Especially the connection with existing work is very well explained which adds to the significance of this paper.

It would be useful if the authors could add a citation showing another example where harmonic mean was used to obtain a smooth objective function. It is a nice transformation.

I am not sure if section 4 should not be restructured. Perhaps it would be better to emphasize the connection with Eq. 5. This could be done if the observation in section 4 was replaced by a theorem. This is an important part, and a clearly explained connection would be nice here.

An important question, how does your a 2-factor approximation gap relate to the problem of over-fitting. It looks that over-fitting is somehow ignored in this work until feature selection is considered? Could you please explain?

The connection with the group lasso regulariser applied to the standard linear regression problem is very interesting.

Practical implications of this work are not clear at this point, but the amount of analytical contribution in this paper makes me believe that this could be a great paper if accepted.
Summary: A strong paper with important and interesting theoretical contributions. Direct practical implications are not clear at this point, but analytical contributions and connections with existing methods are worth accepting this paper.

Author Feedback
Author rebuttal: We would like to thank reviewers for their invaluable comments and suggestions. Below is our response to the concerns raised by the reviewers.

***
Q: Is computing the matrix Q possible in large dimensional applications? (This question is the main concern of reviewer 5 and reviewer 6)

A: To answer this question, we refer the reviewers to section 5 of the submitted manuscript where we demonstrate that, the optimization problem (3) is equivalent to solving a linear regression problem over indicator variables. This fact has also been stated in Remark 1 of the recent work [1] which serves as basis to our computational framework. Based on this linear regression reformulation, computing the matrix Q is not necessary for calculating the classifier.

In order to demonstrate the computational efficiency of our method, we will include the result of an additional numerical experiment on a larger size dataset regarding the GWAS problem with 20,000 features. The numerical experiments on HapMap data shows 6% classification error rate for SVM and 3% classification error rate for the proposed method when combined with the proposed feature selection. The results are averaged over 10 runs.

***
Q: Why do not we compare with SVM?

A: In the initial submitted manuscript, we only compared the proposed algorithm to the methods in the literature using the same framework, i.e., those methods developed based on the marginal distributions. However, due to reviewers' concerns regarding the practicality of the proposed algorithm, we will include the following comparison of SVM and $\delta_{FS,2}^{map}$ method.
Classification Error Rate:
Adult: SVM: 22%, proposed: 16%
Credit: SVM: 16%, proposed: 16%
Kr-vs-kp: SVM: 3%, proposed: 5%
Promoters: SVM: 9%, proposed: 3%
Votes: SVM: 5%, proposed: 2%

In addition to the classification performance table, we will include running times of the classifier in the manuscript and the Matlab code will be publicly available online.

***
Q: Including additional numerical experiments on feature selection method?

A: In addition to the existing simulations, in the revised manuscript, we will include the above GWAS experiment where we select less than 500 features at each experiment. Furthermore, we will include the comparison with the regularized logistic regression on synthetic data where the target variable $y = sign(a^T x + Gaussian noise)$ and $a$ is a sparse vector. The false positives/negatives are reported in discovering the active features for various size problems (d=50-5000, n = 200-20000). The proposed method has marginally outperformed the logistic regression (lassoglm Matlab package) while it has significant computational advantage due to its (group) lasso reformulation. Details are eliminated due to space limitation.

***
Q: Is matrix Q positive semi-definite?

A: Yes. It can be seen that $Q = \mathbf{W} \mathbf{W}^T$ where $\mathbf{W}$ is the random vector defined in section 5. Furthermore, when Q is estimated through the empirical average of data points, we can compute it by $Q = \frac{1}{n} \sum_i (w^i) (w^i)^T$, which is clearly positive semidefinite.

***
Q: How the overfitting issue is addressed in this work?

A: The first source of overfitting in our minimax framework is in estimating the marginals in the set $\mathcal{C}$. This overfitting issue is addressed in section 5 by relating our classification to stochastic optimization and the sample average approximation (SAA) method. As discussed in the paper, one can control the approximation error in SAA method by limiting the search space for $z^*$ or through using ridge regression.
As correctly mentioned by the reviewer, another source of overfitting is the large dimension of the class $\mathcal{C}$. This overfitting could be avoided using the suggested group lasso feature selection approach.

***
Q: The submitted manuscript claims that the gap between (13) and the feature selection algorithm is zero. How is this possible?

A: We do not claim that this gap is zero. As stated in the submitted manuscript, we use the standard group lasso regularization technique. As known in the literature, the essence of this technique is to relax the L_0 norm to the convex L_1 norm. To clarify this point, in the final manuscript, we will explicitly state that (13) is a hard combinatorial problem and the standard (group) lasso technique is used to relax the problem.

***
Q: Is the separability condition the same as standard linear separability in machine learning?

A: No. The separability condition, which is also defined in the recent work [1], is defined for a class of probability distributions and is independent of the classifier at hand. We will clarify this point by providing more details in the revised manuscript.

***
Q: Why are the details of the datasets not given?

A: Since the datasets are publicly available, we prefered not to use more space of the paper for explaining them.